# FedMAP: Meta-Driven Adaptive Differential Privacy for Federated Learning

## Abstract

Federated learning (FL) enables multiple clients to train a shared model without sharing raw data, but gradients can still leak sensitive information through inversion and membership inference attacks. Differential privacy (DP) mitigates this risk by clipping gradients and adding calibrated noise, but most DP-FL methods rely on static noise and clipping schedules. Such rigid designs fail to account for client heterogeneity, changing convergence dynamics, and the growth of cumulative privacy loss. To address these challenges, we propose FEDMAP, a closed-loop framework for adaptive differential privacy in FL. FEDMAP integrates three components. First, a client-side *MetaNet* predicts clipping bounds and noise scales $(C_t, \sigma_t)$ from gradient statistics using a lightweight pretrained BERT-tiny backbone, enabling effective adaptation across communication rounds. Second, a server-side *Rényi DP accountant* tracks heterogeneous privacy costs, computes the global expenditure $\varepsilon_{\text{global}}$, and broadcasts it as a budget signal that constrains cumulative loss and guides client adaptation. Third, a *global feedback regularization* mechanism combines local penalties on per-round privacy cost with global penalties from $\varepsilon_{\text{global}}$, ensuring alignment between client adaptation and the overall budget. Experiments show that FEDMAP improves privacy compliance, and offers stronger robustness against attacks compared with baselines.

## 1 Introduction

Federated learning (FL) (McMahan et al., 2017; Nguyen et al., 2021) allows multiple clients to collaboratively train a shared model without directly sharing their local data. This paradigm is important in domains such as healthcare, speech processing, and mobile sensing, where raw data are highly sensitive (Yu et al., 2020; Khan et al., 2021; Cui et al., 2021). Although FL reduces data exposure, it does not prevent information leakage from shared updates. Recent work shows that adversaries can reconstruct private data through gradient inversion (Zhu et al., 2019; Geiping et al., 2020; Yin et al., 2021) or infer whether samples were used in training through

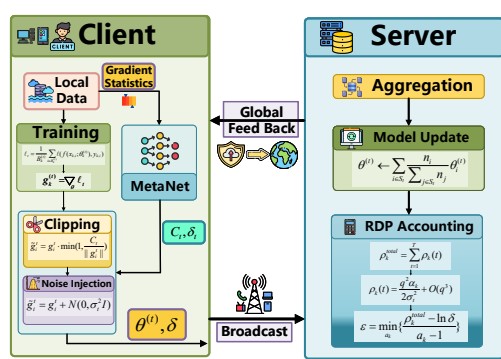

Figure 1: Design of FEDMAP Framework.

membership inference attacks (MIA) (Shokri et al., 2017; Bertran et al., 2023; Thaker et al., 2025). These threats highlight the need for stronger privacy mechanisms in FL.

Differential privacy (DP) (Dwork & Roth, 2014) is a standard approach to protect client data by clipping gradients and adding calibrated noise. DP-SGD (Abadi et al., 2016) and DP-FedAvg (McMahan et al., 2018) provide formal guarantees by applying fixed clipping bounds and noise scales. However, static schedules are not well suited for real federated environments. They ignore client heterogeneity, changing convergence rates, and evolving attack surfaces. As a result, they often trade off model utility against privacy in an inefficient way. Some recent studies propose adaptive or nonuniform budget allocation (Li et al., 2022; Kiani et al., 2025), but these approaches are largely heuristic, centrally managed, and unable to capture real-time dynamics at the client level. Moreover, existing

methods provide limited control over how cumulative privacy loss evolves, which makes long-term budget regulation difficult.

To address these challenges, we propose FEDMAP (**Fed**erated **M**eta-driven **A**daptive **P**rivacy), a closed-loop framework for adaptive differential privacy in FL. As shown in Figure 1, FEDMAP integrates three key components. First, a client-side *MetaNet* predicts clipping bounds and noise scales $(C_t, \sigma_t)$ from gradient statistics. MetaNet employs a lightweight BERT-tiny backbone with frozen intermediate layers and is pretrained on synthetic gradient sequences, which equips it with prior knowledge and improves stability in the early rounds of training. Second, a server-side *Rényi DP accountant* accurately tracks heterogeneous privacy costs across rounds and converts them into user-level $(\varepsilon, \delta)$ guarantees. The accountant computes the global expenditure $\varepsilon_{\text{global}}$ from a lookup table and broadcasts it as a budget signal that constrains cumulative privacy loss and guides client adaptation of $(C_t, \sigma_t)$ in subsequent rounds. Third, a *global feedback regularization* mechanism couples local penalties on per-round privacy cost with global penalties derived from $\varepsilon_{\text{global}}$, ensuring that client-side adaptation remains aligned with the overall budget. This closed-loop design allows clients to dynamically calibrate $(C_t, \sigma_t)$ according to both their local training dynamics and the global privacy constraint, thereby improving the utility-privacy trade-off.

Our main contributions in this work are as follows: (1) We propose FEDMAP, a unified framework for adaptive and client-specific differential privacy in federated learning. (2) We design a **MetaNet** module that uses pretrained BERT-tiny with frozen layers to map gradient statistics into clipping and noise parameters, enabling effective adaptation across communication rounds. (3) We introduce a **global feedback regularization** mechanism that integrates client-side penalties with server-side feedback, forming a closed-loop controller for budget-aware training. (4) We develop a scalable **Rényi DP accountant** that supports heterogeneous noise schedules, maintains negligible runtime overhead, and provides accurate user-level privacy guarantees. (5) We conduct extensive experiments on standard FL benchmarks, showing that FEDMAP improves model accuracy, convergence stability, and privacy compliance, and provides stronger protection against gradient inversion and unseen-class membership inference attacks compared with existing baselines.

## 2 BACKGROUND AND MOTIVATION

Federated learning is a distributed paradigm where clients $\{1, \ldots, N\}$ train a global model on their private datasets $\{\mathcal{D}_i\}$ and send updates to a central server for aggregation. The objective is:

$$\min_{\boldsymbol{\theta}} \mathcal{L}(\boldsymbol{\theta}, \mathcal{D}) = \sum_{i=1}^{N} \frac{|\mathcal{D}_i|}{|\mathcal{D}|} \mathcal{L}_i(\boldsymbol{\theta}, \mathcal{D}_i), \tag{1}$$

where $\boldsymbol{\theta}$ denotes the global model parameters, and $\mathcal{D} = \cup_{i=1}^{N} \mathcal{D}_i$ is the union of all local datasets. A widely used method in this setting is FedAvg (McMahan et al., 2017), which combines local training, client sampling, and model aggregation over multiple communication rounds. To protect privacy, differential privacy (Dwork & Roth, 2014) is commonly applied. A randomized mechanism $\mathcal{M}$ satisfies $(\varepsilon, \delta)$-DP if, for any neighboring datasets $D$ and $D'$ that differ in one record, and for any measurable set $S$, it holds that:

$$\Pr[\mathcal{M}(D) \in S] \leq e^{\varepsilon} \Pr[\mathcal{M}(D') \in S] + \delta, \tag{2}$$

where $\varepsilon$ measures the privacy guarantee and $\delta$ is the failure probability. A standard method is the Gaussian mechanism, which releases $f(D) + \mathcal{N}(0, \sigma^2 \mathbf{I}_d)$, where the noise scale $\sigma$ depends on the function's $\ell_2$-sensitivity $\Delta_2 f$:

$$\sigma \geq \frac{\Delta_2 f}{\varepsilon} \sqrt{2 \ln\left(\frac{1.25}{\delta}\right)}. \tag{3}$$

In FL, local differential privacy (LDP) is enforced by perturbing client updates before transmission. Methods such as DP-SGD (Abadi et al., 2016) and DP-FedAvg (McMahan et al., 2018) add fixed noise to clipped gradients. While theoretically sound, static privacy parameters fail to adapt to client heterogeneity or changing dynamics, often leading to reduced accuracy or weak guarantees.

Recent work has explored dynamic privacy strategies. For example, DPSFL adjusts clipping thresholds over time (Zhang et al., 2024), a time adaptive budget allocation scheme has been introduced (Kiani et al., 2025), and a wavelet based perturbation mechanism distributes noise based on

gradient structure (Ranaweera et al., 2025). These methods show that adaptive strategies improve the balance between privacy and utility. Another challenge is information leakage from gradients, as gradient inversion attacks (Zhu et al., 2019; Geiping et al., 2020) reveal that large gradient entries may expose sensitive features of local data. To mitigate this risk, prior work has studied gradient clipping and perturbation, showing that targeted noise injection and clipping-aware designs improve robustness while preserving utility (Fu et al., 2021; Zhang et al., 2022). Beyond gradient inversion, membership inference attacks pose an additional risk. Classical MIA rely on prediction confidence to distinguish training from nontraining samples, while recent work shows that quantile-based MIA remain effective even in the *unseen-class* setting (Bertran et al., 2023; Thaker et al., 2025). These findings confirm that FL models face risks from both gradient inversion and membership inference, motivating the design of adaptive and budget-aware DP mechanisms.

Despite these advances, most existing methods still rely on globally fixed or heuristic privacy schedules. Such strategies lack flexibility to handle heterogeneous client data, changing convergence patterns, and evolving attack risks across training. They also do not regulate the cumulative privacy budget over time, making it difficult to balance model utility with long-term privacy. To overcome these limitations, we propose FEDMAP.

## 3   PROPOSED DESIGN OF FEDMAP

This design is a closed-loop framework that consists of three core components. (i) A client-side *MetaNet* adaptively generates clipping bounds and noise scales. (ii) A server-side *Rényi DP accountant* records and updates the privacy cost in each round. (iii) A *global feedback regularization* aligns local adaptation with the global privacy budget. In each round, the client uses MetaNet to produce $(C_t, \sigma_t)$, applies DP-SGD, and sends the *clipped-and-noised* updates together with $\sigma_t$, which is required for accounting. The server aggregates heterogeneous costs through RDP and converts them into user-level $(\varepsilon, \delta)$. The resulting $\varepsilon_{\text{global}}$ is then broadcast to the clients participating in the next round of training, guiding the next choice of $(C_{t+1}, \sigma_{t+1})$.

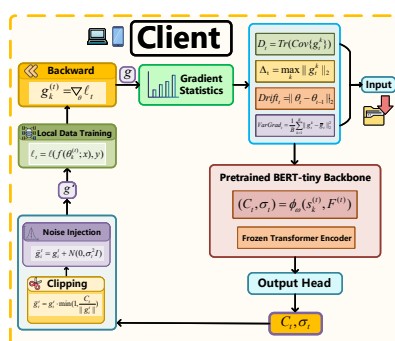

Figure 2: Structure of MetaNet.

### 3.1   METANET FOR ADAPTIVE DP CALIBRATION

As shown in Figure 2, the meta-network is the core module of FEDMAP. Its task is to adaptively generate the clipping bound $C_t$ and the noise scale $\sigma_t$ in each training round. Fixed strategies with static parameters cannot handle heterogeneous data distributions or changing convergence behavior. They often lead to either a large loss in utility or weak privacy protection. MetaNet addresses this issue by adjusting privacy parameters in a data-driven and adaptive way, which preserves the balance between model utility and privacy guarantees.

**Gradient Statistics and Input Features**. Formally, MetaNet is a lightweight client-side network. It takes as input a compact feature vector $h_t \in \mathbb{R}^4$, which is derived from four gradient-based statistics:

$$h_t = \big(D_t, \Delta_t, \text{Drift}_t, \text{VarGrad}_t\big). \tag{4}$$

These statistics are defined as:

$$D_t = \text{Tr}\big(\text{Cov}\{g_k^t\}_{k \in \text{mini-batch}}\big), \quad \Delta_t = \max_k \|g_k^t\|_2,$$

$$\text{Drift}_t = \|\theta_t - \theta_{t-1}\|_2, \quad \text{VarGrad}_t = \frac{1}{B} \sum_{k=1}^{B} \|g_k^t - \bar{g}_t\|_2^2.$$

Here $g_k^t$ is the gradient of sample $k$ at round $t$, $\bar{g}_t$ is the mini-batch mean gradient, and $\theta_t$ is the local model parameter. $\Delta_t$ relates to sensitivity and clipping bias. $D_t$ and VarGrad$_t$ capture dispersion that drives stochastic and DP variance. Drift$_t$ reflects nonstationarity that affects stable choices of

$(C_t, \sigma_t)$. Prior work (Andrew et al., 2021; Fu et al., 2022; Allouah et al., 2025) has confirmed the effectiveness of such features for adaptive clipping and noise scheduling.

**Backbone Design and Pretraining Strategy**. Although MetaNet is expressive, it lacks prior knowledge, which may cause unstable predictions in early training rounds. To address this issue, we adopt a lightweight BERT-tiny backbone. It consists of two Transformer encoder layers with hidden size $d_h = 64$ and two attention heads. The model is initialized with pretrained weights and fine-tuned for the task to capture temporal dynamics. Gradient statistics from consecutive rounds are mapped into the Transformer input space, and the [CLS] embedding is used to predict $(C_t, \sigma_t)$. Following prior findings in transfer learning (Lee et al., 2019; Grießhaber et al., 2020; Liu et al., 2021), we apply a frozen-middle-layer strategy. Only the input projection and the output head are updated during fine-tuning. This reduces computation and enables effective adaptation to privacy dynamics.

To further mitigate cold-start effects, we first fine-tune the model on synthetic sequences of gradient statistics. Empirical distributions of the four features are estimated from federated training runs on datasets such as CIFAR-10, FashionMNIST, and SVHN. Synthetic sequences are then generated by log-normal sampling. Labels for $(C, \sigma)$ are constructed from predefined feature-to-parameter mappings. For example, the clipping bound $C$ is set proportional to the empirical mean plus variance of gradient norms, while the noise scale $\sigma$ is set proportional to feature variance. We use sequences of length 5 to train the model. The training objective is the mean squared error:

$$\mathcal{L}_{\text{MSE}} = \|(C, \sigma)_{\text{pred}} - (C, \sigma)_{\text{label}}\|_2^2, \tag{5}$$

and optimization is done with Adam at learning rate $1 \times 10^{-4}$. Gradient clipping with max-norm 1.0 and early stopping are applied. Only the projection and output layers are updated, while the Transformer layers remain frozen. This synthetic pretraining equips the model with prior knowledge and enables reliable predictions in early training. Experiments show that it generalizes well across datasets and captures the mapping between gradient features and privacy parameters effectively, which provides empirical stability.

**Takeaways 3.1.** *Each client keeps its own MetaNet instance for personalized $(C_t, \sigma_t)$ under heterogeneous data. This improves fairness and stability compared with global static schedules. MetaNet is lightweight and stable. Input dimension is 4, hidden size is 64, depth is 2 ($\approx 0.2M$ parameters). Thus, inference is lightweight, and latency is within 1ms.*

## 3.2 GLOBAL PRIVACY ACCOUNTING UNDER DYNAMIC NOISE

While client-side MetaNet adaptively calibrates local clipping and noise, rigorous user-level privacy still requires accurate global accounting across training rounds. In federated learning, each client contributes partially perturbed (clipped-and-noised) updates, and the overall $(\varepsilon, \delta)$-DP guarantee depends on the cumulative privacy loss over $T$ rounds. Static accountants assume fixed noise and clipping schedules, which are inconsistent with the adaptive parameters of FEDMAP. To address this, we adopt a server-side Rényi Differential Privacy (RDP) accountant (Mironov, 2017), extended to handle non-uniform and dynamically scheduled Gaussian mechanisms.

**Per-Round RDP Cost and Global Composition**. In each round $t$, client $i$ selects $(C_t^{(i)}, \sigma_t^{(i)})$ through MetaNet, applies DP-SGD locally, and uploads its clipped-and-noised updates together with $\sigma_t^{(i)}$. The server aggregates these updates and computes the effective sampling rate $q = |S_t|/N$, where $S_t$ is the set of participating clients. For brevity, we denote $q$ per round; when participation varies, the expressions hold with $q^2$ replaced by $q_t^2$ and $Tq^3$ by $\sum_t q_t^3$.

We parameterize the Gaussian noise as $\mathcal{N}(0, \sigma_t^2 I)$. where $\sigma_t^{(i)}$ directly denotes the noise standard deviation chosen by MetaNet. Thus, the RDP term depends on $\sigma_t$ and $q$, while sensitivity is controlled by $C_t$. Following the subsampled Gaussian mechanism (WANG et al., 2020), the per-round Rényi divergence of order $\alpha_k$ is approximated by

$$\rho_k^{(i)}(t) = \frac{q^2 \alpha_k}{2\left(\sigma_t^{(i)}\right)^2} + O(q^3), \tag{6}$$

where clipping at $C_t^{(i)}$ ensures bounded sensitivity. This generalizes prior work on non-uniform privacy allocation (Kiani et al., 2025), enabling FEDMAP to handle dynamic noise levels across

heterogeneous clients and rounds. Then, the global privacy cost for client $i$ after $T$ rounds is

$$\rho_k^{(i),\text{total}} = \sum_{t=1}^{T} \rho_k^{(i)}(t). \tag{7}$$

Applying the RDP-to-DP conversion,

$$\varepsilon^{(i)} = \min_{\alpha_k} \left\{ \rho_k^{(i),\text{total}} - \frac{\log \delta}{\alpha_k - 1} \right\}, \tag{8}$$

yields a user-level $(\varepsilon, \delta)$ guarantee. To reduce computational cost, the server maintains a lookup table of $\rho_k(\sigma, q)$ values for a grid of $\sigma$ and $\alpha_k$, and interpolates at runtime. The global expenditure is conservatively defined as

$$\varepsilon_{\text{global}} = \max_i \varepsilon^{(i)}, \tag{9}$$

which is broadcast to clients. This design supports real-time privacy tracking with negligible cost, ensuring scalability in large-scale FL deployments.

**Global Feedback and Budget Regulation**. While server-side RDP accounting provides accurate retrospective guarantees, proactive regulation of privacy requires coupling global feedback with local adaptation. To this end, the cumulative privacy expenditure $\varepsilon_{\text{global}}$ is periodically broadcast to clients as a penalty signal. This feedback encourages MetaNet to adjust $(C_t, \sigma_t)$ in subsequent rounds, preventing premature exhaustion of the privacy budget.

For completeness, we record a practical upper bound under bounded $\sigma_t^{(i)}$ and $q$. If $\sigma_t^{(i)} \in [\sigma_{\min}, \sigma_{\max}]$ and $q \leq q_{\max}$, then for any $\alpha_k > 1$,

$$\varepsilon_{\text{global}} \leq \min_{\alpha_k > 1} \left\{ \frac{\alpha_k}{2\sigma_{\min}^2} \sum_{t=1}^{T} q^2 - \frac{\log \delta}{\alpha_k - 1} \right\} + O\left(Tq^3\right). \tag{10}$$

This upper bound highlights that the global privacy cost grows with both the number of rounds $T$ and the maximum sampling rate $q_{\max}$. It also shows that smaller noise scales $\sigma_{\min}$ lead to larger privacy expenditure, while larger $\alpha_k$ values provide tighter bounds. Thus, Eq. equation 10 provides a conservative yet interpretable estimate of the worst-case privacy loss, which is useful for understanding how adaptive schedules interact with global budget constraints.

**Takeaways 3.2.** *The server maintains a lookup table of $\rho_k(\sigma, q)$ and interpolates at runtime, enabling negligible accounting latency in our implementation even with $N = 10^3$ clients. In practice, we discretize $\sigma$ into 100 bins and $\alpha_k$ into $\{2, 4, 8, 16, 32, 64\}$, which balances accuracy and speed. The broadcasted $\varepsilon_{\text{global}}$ is a single scalar, adding no noticeable communication overhead.*

### 3.3 PRIVACY LOSS REGULARIZATION WITH GLOBAL FEEDBACK

**Local Regularization with Privacy Proxy**. In FEDMAP, each client uses MetaNet to predict its privacy parameters $(C_t, \sigma_t)$ for round $t$, where $C_t$ is the clipping bound and $\sigma_t$ is the DP-SGD noise scale. These parameters directly govern a local proxy of the per-round privacy cost. Regularizing this proxy is essential to avoid erratic spending and to align client behavior with the global budget. By the Gaussian mechanism (Dwork & Roth, 2014), the per-round guarantee is approximated as

$$\varepsilon_t \approx \frac{C_t}{\sigma_t} \sqrt{2 \ln(1.25/\delta)}. \tag{11}$$

In practice, we use the heuristic $\widehat{\varepsilon}_t$ from Eq. equation 11 for client-side regularization. Rigorous user-level guarantees are always provided by server-side RDP accounting in Sec. 3.2, and $\widehat{\varepsilon}_t$ is never used as the reported guarantee. To stabilize adaptation, we impose a quadratic penalty that aligns the predicted $(C_t, \sigma_t)$ with a target budget $\varepsilon_{\text{target}}$:

$$L_{\text{privacy}}^{(t)} = (\widehat{\varepsilon}_t - \varepsilon_{\text{target}})^2. \tag{12}$$

This reduces variance in per-round expenditure and keeps local updates close to the desired operating point. For any window of length $T'$, define the empirical deviation.

$$\gamma_{T'} := \frac{1}{T'} \sum_{t=1}^{T'} \left| \widehat{\varepsilon}_t - \varepsilon_{\text{target}} \right|.$$

Then, by the triangle inequality, we have

$$T'(\varepsilon_{\text{target}} - \gamma_{T'}) \leq \sum_{t=1}^{T'} \widehat{\varepsilon}_t \leq T'(\varepsilon_{\text{target}} + \gamma_{T'}), \tag{13}$$

which bounds the average ratio $\frac{1}{T'}\sum_{t=1}^{T'}(C_t/\sigma_t)$ and thus the average $\frac{1}{T'}\sum_{t=1}^{T'}\sigma_t^{-2}$. Intuitively, this inequality shows that the cumulative local expenditure over any window of length $T'$ cannot drift far from the target budget. The deviation is limited by $\gamma_{T'}$, which quantifies the average mismatch from $\varepsilon_{\text{target}}$. As a result, local penalties keep short-term fluctuations under control, while ensuring long-term expenditure remains consistent with the global privacy bound in Sec. 3.2.

**Global Feedback and Closed-Loop Control**. Local constraints alone are insufficient, since cumulative expenditure over $T$ rounds must also respect user-level guarantees. We therefore introduce a global penalty derived from server-side Rényi DP accounting (Mironov, 2017):

$$L_{\text{global}} = \beta \cdot \varepsilon_{\text{global}}, \tag{14}$$

where $\varepsilon_{\text{global}}$ is the cumulative privacy loss up to the current round (taken as the maximum across users), and $\beta > 0$ controls the strength of global feedback. This term encourages budget-efficient adaptation and prevents premature exhaustion of the global budget.

The two penalties are combined with the prediction loss to form the overall objective:

$$\mathcal{L}_{\text{total}} = \frac{1}{|B|} \sum_{i \in B} \mathcal{L}_i(\theta, x_i) + \lambda L_{\text{privacy}}^{(t)} + L_{\text{global}}, \tag{15}$$

where $\mathcal{L}_i(\cdot)$ is the task loss for client $i$, $\lambda$ balances task utility against local regularization, and $L_{\text{global}}$ provides server-driven feedback. We grid search $\lambda, \beta \in \{0.01, 0.05, 0.1, 0.5, 1.0\}$ and select $\lambda = 0.1, \beta = 0.1$ for all main experiments.

This closed-loop design couples local calibration with global accounting. Clients adaptively tune $(C_t, \sigma_t)$ not only to stabilize per-round expenditure but also to align with the global privacy budget. Theoretically, this reduces variance in cumulative privacy loss and concentrates it around the target budget. Practically, FEDMAP learns budget-aware strategies end-to-end, improving the utility-privacy trade-off under heterogeneous data and enhancing robustness against attacks.

**Takeaways 3.3.** *This scheme resembles a proportional controller: local penalties reduce deviation from per-round targets, while global penalties prevent premature exhaustion of the long-term budget. Both penalties require only scalar computations per round and add negligible overhead.*

### 3.4 THEORETICAL ANALYSIS

This section formalizes how the three components of FEDMAP jointly determine the utility-privacy trade-off. We adopt standard assumptions consistent with prior work (Lian et al., 2018; Nguyen et al., 2022; Fu et al., 2022; Xiong et al., 2024; Wang et al., 2024):

**Assumption 1.** $\mathcal{L}_i(\cdot, x)$ is $L$-smooth in $\theta$ for all $i, x$. Hence, the global objective $\mathcal{L}$ is $L$-smooth.

**Assumption 2.** For a mini-batch of size $B$, per-sample gradients satisfy $\mathbb{E}\|g_k^t\|_2^2 \leq G^2$. The variance of the mini-batch gradient noise is bounded by $\sigma_g^2/B$.

**Assumption 3.** The client sampling rate is $q_t = |S_t|/N \leq q_{\max}$, with $q_{\max} \ll 1$.

Under Assumptions 1–3, adaptive DP-SGD with clipping at $C_t$ and Gaussian noise $\mathcal{N}(0, \sigma_t^2 I)$ yields the following bounds.

**Theorem 1.** *Under Assumptions 1–3, with $\eta_t \leq 1/L$, we have*

$$\frac{1}{T} \sum_{t=1}^{T} \mathbb{E}\|\nabla \mathcal{L}(\theta_t)\|_2^2 \leq \frac{2(\mathcal{L}(\theta_1) - \mathcal{L}^\star)}{T\eta_T} + \frac{2}{T} \sum_{t=1}^{T} \left( L\|b_t\|_2^2 + L\eta_t(\frac{\sigma_g^2}{B} + d\sigma_t^2) \right), \tag{16}$$

*where $b_t = \mathbb{E}[\frac{1}{B}\sum_{k=1}^{B} \text{clip}(g_k^t, C_t)] - \nabla \mathcal{L}(\theta_t)$.*

**Theorem 2.** *If $\mathcal{L}$ is $\mu$-strongly convex and $\eta \leq 1/(2L)$ is constant, then after $T$ rounds*

$$\mathbb{E}[\mathcal{L}(\theta_T) - \mathcal{L}^\star] \leq (1 - \mu\eta)^T (\mathcal{L}(\theta_0) - \mathcal{L}^\star) + \frac{\eta L}{\mu} \cdot \frac{1}{T} \sum_{t=1}^{T} \left( \|b_t\|_2^2 + \eta \frac{\sigma_g^2}{B} + d\sigma_t^2 \right). \tag{17}$$

The right-hand sides of Theorems 1–2 are dominated per round by two contributions: (i) a clipping *bias* term $\|b_t(C_t)\|_2^2$; and (ii) a *variance* term from stochastic sampling $\sigma_g^2/B$ and DP noise with standard deviation $\sigma_t$. Collecting these with their weights $L$ and $\eta_t$ motivates the following surrogate for choosing $(C_t, \sigma_t)$:

$$\mathcal{U}_t(C_t, \sigma_t) := L\|b_t(C_t)\|_2^2 + L\eta_t\left(\frac{\sigma_g^2}{B} + d\sigma_t^2\right). \tag{18}$$

Reducing $\mathcal{U}_t$ tightens the bounds in Theorems 1–2 by balancing clipping bias, which decreases with larger $C_t$, against DP variance, which increases with $\sigma_t^2$. This connects the feature design in Sec. 3.1 to optimization error, where $\Delta_t$ reflects clipping bias and $D_t$ and $\mathrm{VarGrad}_t$ capture variance.

## 4 EXPERIMENTS

### 4.1 EXPERIMENTAL SETUP

**Datasets and Models.** We evaluate the effectiveness of our defense mechanisms against gradient inversion attacks using three publicly available datasets, CIFAR-10 (Krizhevsky et al., 2009), FashionMNIST (Xiao et al., 2017), and SVHN (Yuval, 2011), with a mini-batch size of 4. We use the same datasets to assess reconstruction quality and resistance to unseen-class membership inference attacks. CIFAR-10 and SVHN are paired with a randomly initialized ResNet-18 (He et al., 2016), while FashionMNIST is trained with LeNet (LeCun et al., 1998). To evaluate training dynamics under privacy protection, we further conduct experiments on these datasets . All datasets are partitioned across $N = 100$ clients using a Dirichlet distribution with concentration parameter $\alpha$, which controls the degree of statistical heterogeneity. Smaller $\alpha$ values yield more skewed, non-IID data distributions. Following prior work (Wang et al., 2020; Dai et al., 2022; Cao & Gong, 2022; Oh et al., 2022; Chen & Chao, 2022), we set $\alpha = 0.5$ unless otherwise specified.

**Training Settings.** We follow a standard federated setup with 10 randomly selected clients out of 100 per round. Local training uses a batch size of 64 and SGD optimizer, with dataset specific settings: three epochs at $\eta = 0.01$ for FashionMNIST, eight epochs at $\eta = 0.001$ for CIFAR-10, and five epochs at $\eta = 0.1$ for SVHN. To ensure reproducibility, all experiments are performed with a fixed random seed (42) and executed on NVIDIA RTX 5090 GPUs.

**Attack Baselines.** We evaluate FEDMAP against two types of privacy attacks. For membership inference, we adopt *Logistic Regression MIA* (Shokri et al., 2017), a shadow model based method that trains a binary classifier on confidence features, and *Quantile-MIA* (Bertran et al., 2023), a shadow model free black box attack that leverages quantile regression, which has been shown to be more effective in the unseen-class setting (Thaker et al., 2025). For gradient inversion, we consider *DLG* (Zhu et al., 2019), which reconstructs inputs by minimizing the gradient distance, *IG* (Geiping et al., 2020), which maximizes gradient cosine similarity, and *GI* (Yin et al., 2021), which initializes from Gaussian noise and iteratively refines reconstructions using gradient and model statistics.

**Defense Baselines.** To provide a comprehensive comparison, we include two state-of-the-art defenses. *Soteria* (Sun et al., 2021) reduces privacy leakage by pruning gradients in fully connected layers using a sensitivity mask that filters out high-risk components. *CENSOR* (Zhang et al., 2025) protects private information by projecting gradients into a subspace orthogonal to the inferred attack space through Bayesian sampling, thereby mitigating leakage while maintaining model performance.

**Evaluation Metrics.** We adopt three standard metrics to quantify reconstruction quality and privacy leakage. *Mean Squared Error (MSE)* measures pixel-wise differences between original and reconstructed images; higher values indicate stronger privacy. *Peak Signal-to-Noise Ratio (PSNR)* quantifies signal clarity relative to noise; lower values suggest greater distortion and better protection. *Structural Similarity Index Measure (SSIM)* evaluates perceptual similarity based on luminance, contrast, and structure; lower scores reflect reduced visual resemblance and improved privacy. For membership inference evaluation, we report true positive rates at low false positive regimes: *TPR@1%FPR* and *TPR@0.1%FPR*, which reflect the attack success rate when only a small fraction of non-members are misclassified. Lower values indicate stronger resilience against MIA.

Table 1: Defense performance comparison on SVHN dataset under different attacks.

| Dataset | Defense Method | DLG | | | IG | | | GI | | |
|---|---|---|---|---|---|---|---|---|---|---|
| | | MSE ↓ | PSNR ↑ | SSIM ↑ | MSE ↓ | PSNR ↑ | SSIM ↑ | MSE ↓ | PSNR ↑ | SSIM ↑ |
| SVHN | None | 0.0006 | 39.29 | 0.9929 | 0.0150 | 34.49 | 0.8858 | 0.0074 | 35.66 | 0.8986 |
| | DP | 0.0743 | 11.43 | 0.0597 | 0.0690 | 11.73 | 0.1003 | 0.0695 | 11.70 | 0.0739 |
| | Soteria | 0.0711 | 11.67 | 0.0790 | 0.0647 | 12.62 | 0.2363 | 0.0275 | 22.21 | 0.6909 |
| | CENSOR | 0.0728 | 11.54 | 0.0505 | 0.0740 | 11.46 | 0.0999 | 0.0753 | 11.44 | 0.0768 |
| | FEDMAP | 0.0796 | 11.24 | 0.0594 | 0.0761 | 11.39 | 0.1186 | 0.0786 | 11.24 | 0.0622 |
| CIFAR-10 | None | 0.0091 | 29.88 | 0.8633 | 0.0337 | 21.70 | 0.5461 | 0.0388 | 17.96 | 0.4482 |
| | DP | 0.0679 | 11.78 | 0.1060 | 0.0645 | 12.06 | 0.1221 | 0.0676 | 11.83 | 0.0756 |
| | Soteria | 0.0603 | 12.59 | 0.1524 | 0.0558 | 12.99 | 0.1828 | 0.0419 | 17.70 | 0.4606 |
| | CENSOR | 0.0682 | 11.82 | 0.0466 | 0.0770 | 11.51 | 0.1141 | 0.0665 | 11.93 | 0.0783 |
| | FEDMAP | 0.0755 | 11.54 | 0.0483 | 0.0730 | 11.57 | 0.0892 | 0.0685 | 11.85 | 0.0704 |
| FMNIST | None | 0.1455 | 8.45 | 0.0688 | 0.1453 | 8.41 | 0.0666 | 0.1694 | 7.86 | 0.0653 |
| | DP | 0.1577 | 8.09 | 0.0246 | 0.1556 | 8.14 | 0.0327 | 0.1653 | 7.92 | 0.0390 |
| | Soteria | 0.1518 | 8.23 | 0.0846 | 0.1412 | 8.59 | 0.0869 | 0.1664 | 7.87 | 0.0867 |
| | CENSOR | 0.1518 | 8.22 | 0.0279 | 0.1537 | 8.16 | 0.0287 | 0.1701 | 7.75 | 0.0238 |
| | FEDMAP | 0.1641 | 7.90 | 0.0342 | 0.1575 | 8.07 | 0.0267 | 0.1709 | 7.75 | 0.0415 |

Table 2: MIA defense performance comparison across different attacks and datasets.

| Dataset | Defense | Logistic Regression MIA | | Quantile MIA | |
|---|---|---|---|---|---|
| | | TPR@1%FPR | TPR@0.1%FPR | TPR@1%FPR | TPR@0.1%FPR |
| SVHN | None | 26.50% | 15.20% | 9.55% | 1.67% |
| | DP | 20.03% | 10.01% | 7.11% | 0.97% |
| | Soteria | 17.89% | 8.51% | 6.54% | 0.53% |
| | CENSOR | 17.57% | 11.80% | 6.26% | 1.20% |
| | FEDMAP | **13.59%** | **6.22%** | **4.65%** | **0.18%** |
| CIFAR-10 | None | 1.91% | 0.44% | 8.65% | 2.44% |
| | DP | 0.55% | 0.11% | 7.01% | 1.41% |
| | Soteria | 0.87% | 0.21% | **5.20%** | 1.02% |
| | CENSOR | 0.75% | 0.23% | 6.42% | 1.87% |
| | FEDMAP | **0.36%** | **0.06%** | 5.80% | **0.89%** |
| FMNIST | None | 20.65% | 14.95% | 8.06% | 0.94% |
| | DP | 16.46% | 12.48% | 2.86% | 0.71% |
| | Soteria | 18.03% | 11.53% | 2.15% | 0.50% |
| | CENSOR | 16.78% | 13.02% | 2.69% | 0.62% |
| | FEDMAP | **13.06%** | **5.08%** | **1.48%** | **0.29%** |

## 4.2 EXPERIMENTAL RESULTS

**Main Results.** Table 1 shows the performance of DLG, IG, and GI under different defenses on SVHN, CIFAR-10, and FashionMNIST. We follow prior work (Zhu et al., 2019; Geiping et al., 2020; Yin et al., 2021) and evaluate privacy leakage in the first communication round, which is the most vulnerable stage. This setup provides a strong proxy for worst case exposure. Across all datasets, unprotected models yield accurate reconstructions with low MSE and high SSIM. Once defenses are applied, attack success drops sharply. FEDMAP achieves robust protection and is often stronger than Soteria and CENSOR. For example, on CIFAR-10 with DLG, FEDMAP lowers SSIM to 0.0483 compared to 0.8633 without defense. On SVHN and FashionMNIST, FEDMAP also maintains competitive MSE and SSIM while suppressing reconstruction fidelity more effectively than standard DP. While Table 2 reports results on unseen-class MIA, where two classes are excluded from training and used only as attack targets. Unprotected models show severe leakage, with TPR above 26% on SVHN. Standard defenses reduce attack accuracy, but FEDMAP provides the most reliable protection. On CIFAR-10, it lowers TPR@0.1%FPR to 0.06%, and on SVHN to 0.18%, both substantially better than DP, Soteria, and CENSOR. These results demonstrate that FEDMAP offers adaptive and robust defense against gradient inversion and unseen-class membership inference, outperforming existing baselines under strong privacy threats.

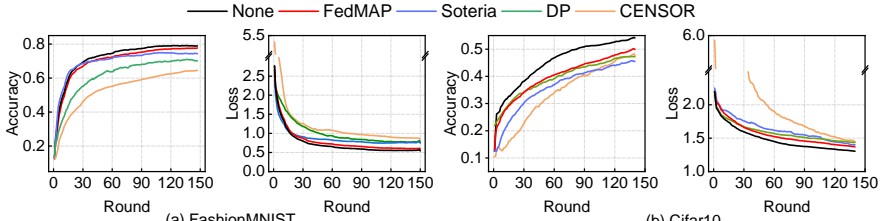

Figure 3: Training loss and test across communication rounds.

**Convergence Evaluation.** Figure 3 shows the convergence of different defenses on FashionMNIST and CIFAR-10. The unprotected model serves as an upper bound. FedMAP converges faster and more stably than DP, Soteria, and CENSOR. On FashionMNIST, it quickly approaches the non-private baseline with low loss, while Soteria and DP converge slowly and CENSOR remains unstable. On CIFAR-10, FedMAP again achieves higher accuracy and lower loss, while the other defenses lag behind. These results show that FedMAP provides a favorable balance between utility and privacy, delivering superior convergence and final performance in federated learning.

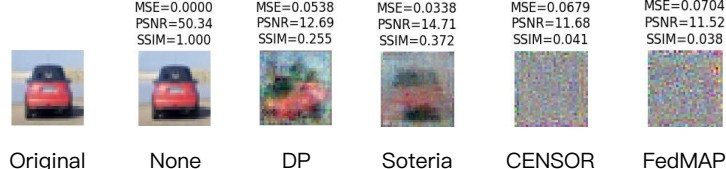

Figure 4: Visual comparison of reconstructed images under the DLG attack.

**Qualitative Evaluation.** Figure 4 shows reconstructed images under different defenses against the DLG attack on CIFAR-10. Without protection, the original image can be almost perfectly recovered, which poses a severe privacy risk. DP and Soteria introduce distortion but still reveal recognizable structures, showing incomplete protection. CENSOR increases distortion but generates images with strong noise artifacts. In contrast, FedMAP produces heavily degraded reconstructions with the lowest SSIM and comparable distortion metrics, where semantic details of the original image are completely lost. These results show that FedMAP provides the strongest resistance to gradient inversion and greatly reduces the risk of information leakage.

Table 3: Sensitivity Analysis Results of Hyperparameters $\lambda$ and $\beta$

| $\lambda$ | 0.01 | 0.1 | 0.5 | 1.0 | $\beta$ | 0.01 | 0.1 | 0.5 | 1.0 |
|---|---|---|---|---|---|---|---|---|---|
| acc | 0.4655 | 0.4721 | 0.4649 | 0.4632 | acc | 0.4693 | 0.4721 | 0.4679 | 0.4633 |
| $\varepsilon$ | 5.153 | 4.992 | 4.973 | 4.959 | $\varepsilon$ | 5.114 | 4.992 | 4.982 | 4.960 |

**Sensitivity to $\lambda$ and $\beta$.** Table 3 shows that increasing either $\lambda$ or $\beta$ slightly reduces accuracy but lowers the cumulative privacy loss $\varepsilon$. This confirms their role in balancing utility and privacy. Larger coefficients give stricter control of privacy leakage at the cost of marginal utility degradation. Smaller values relax the constraints, improving accuracy but weakening privacy protection.

**Ablation Study.** Table 4 shows that FedMAP maintains a favorable balance between accuracy and privacy loss across different client participation rates $C$. On FashionMNIST, increasing $C$ from 0.1 to 0.3 raises accuracy from 0.7824 to 0.8011 with a moderate increase of $\epsilon$, indicating that FedMAP can use more client updates without sharply degrading privacy. A similar trend is observed on CIFAR-10 and

Table 4: Performance under different client participation rates $C$.

| $C$ | Fashion-MNIST | | CIFAR-10 | | SVHN | |
|---|---|---|---|---|---|---|
| | acc | $\epsilon$ | acc | $\epsilon$ | acc | $\epsilon$ |
| 0.1 | 0.7824 | 1.04 | 0.4253 | 2.19 | 0.4016 | 2.23 |
| 0.2 | 0.7940 | 3.57 | 0.4435 | 8.81 | 0.4813 | 10.38 |
| 0.3 | 0.8011 | 7.71 | 0.4700 | 19.78 | 0.4926 | 27.64 |

SVHN, where accuracy improves steadily as $C$ increases and $\epsilon$ grows in a predictable way. These results show that FedMAP adapts flexibly to varying participation rates while preserving utility and keeping privacy budgets under control.

## 5 CONCLUSIONS AND LIMITATIONS

In conclusion, we present FEDMAP, a closed-loop framework for adaptive differential privacy in federated learning. It integrates three components: a client-side MetaNet that predicts clipping bounds and noise scales, a server-side Rényi DP accountant that computes the global expenditure $\varepsilon_{\text{global}}$, and a global feedback regularization mechanism that aligns local adaptation with the overall budget. Together, these modules enable dynamic calibration of privacy parameters, improve the utility-privacy trade-off, and enhance robustness against attacks. Nevertheless, FEDMAP relies on synthetic pretraining for MetaNet, which may limit generalization to real workloads, and its evaluation focuses on image classification benchmarks. Future work will extend FEDMAP to larger and more diverse FL settings, and explore stronger theoretical guarantees and online adaptation strategies to reduce reliance on synthetic pretraining.

## REPRODUCIBILITY STATEMENT

We have made extensive efforts to ensure the reproducibility of our work. The main paper provides detailed descriptions of the proposed algorithm, model architectures, training setup, hyperparameter configurations, data preprocessing steps, and theoretical results. If the paper is accepted, we will release the code on GitHub. During the rebuttal phase, if reviewers request to check the relevant code, we will upload it to an anonymous GitHub repository to facilitate the review process.

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

# A  APPENDIX

## A.1  SYSTEM WORKFLOW

The complete workflow of the proposed FEDMAP framework is provided in Algorithm 1. FEDMAP builds upon the standard FedAvg protocol and incorporates three core components that collectively support efficient, personalized, and privacy-preserving training.

At the beginning of each round $t$, the server randomly selects a subset of clients $S_t$ according to a predefined sampling ratio $p$, and broadcasts the current global model $\theta^{(0)}$ to the selected clients. Each client initializes its local model by setting $\theta_i^{(t)} \leftarrow \theta^{(t-1)}$ and starts the PretrainedBertMetaNet model if it has not already been initialized. During local training, each client performs $E$ epochs over its local dataset. For each mini-batch, the client computes the forward loss and the corresponding

---

**Algorithm 1** FEDMAP Framework

---

**Require:** Client set $\mathcal{N} = \{1, ..., N\}$, sampling ratio $p$, communication rounds $T$, local epochs $E$, batch size $B$, Fine-tuned **MetaNet** model
**Ensure:** Final global model $\theta^{(t)}$
1: Initialize global model $\theta^{(0)}$ and get $K_s \leftarrow \max(p \cdot N, 1)$
2: **for** $t = 1$ to $T$ **do**
3:      Randomly select clients $S_t$ of size $K_s$
4:      Server broadcasts global model $\theta^{(t-1)}$ and current penalty $\epsilon_{\text{global}}$ to each $i \in S_t$
5:      **for all** clients $i \in S_t$ **in parallel do**
6:          Set local model $\theta_i^{(t)} \leftarrow \theta^{(t-1)}$ and load Fine-tuned **MetaNet**
7:          **for** $e = 1$ to $E$ **do**
8:              **for** each batch $b \in D_i$ **do**
9:                  Compute forward loss $\ell_i = \ell(f(\theta_i^{(t)}; x), y)$
10:               Perform backward pass to obtain gradients and related statistics
11:               Extract gradient-related features and input them into **MetaNet**
12:               Obtain new privacy parameters $(C_i^{(t)}, \sigma_i^{(t)})$ from **MetaNet**
13:               Clip gradients and inject Gaussian noise to obtain $g_{\text{noisy}}$
14:               Update local model parameters $\theta_i^{(t)} \leftarrow \theta_i^{(t)} - \eta \cdot g_{\text{noisy}}$
15:              **end for**
16:          **end for**
17:          Upload $\theta_i^{(t)}$ and effective noise scale $\sigma_i^{(t)}$ to the server
18:      **end for**
19:      Server aggregates updates:

$$\theta^{(t)} \leftarrow \sum_{i \in S_t} \frac{n_i}{\sum_{j \in S_t} n_j} \theta_i^{(t)} \tag{20}$$

20:      Server performs RDP privacy accounting using $\{\sigma_i^{(t)}, |S_t|, p\}$
21:      Update cumulative privacy budget: $\epsilon_{\text{global}}$
22: **end for**

---

gradient via backpropagation. The current training state, including the gradient diversity, sensitivity proxy, parameter drift and gradient variance, is passed into the PretrainedBertMetaNet. It then returns updated privacy parameters $(C_i^{(t)}, \sigma_i^{(t)})$, which determine the clipping bound and the scale of noise to be added.

The local gradient is first clipped to ensure bounded sensitivity and then perturbed using the noise injection strategy. Specifically, Gaussian noise with a standard deviation of $\sigma_t$ is injected into the updated gradients. The estimated privacy cost $\epsilon_t$ is computed from $(C_i^{(t)}, \sigma_i^{(t)})$. A privacy-loss regularization term $L_{\text{privacy}}^{(t)}$, augmented with global feedback, is incorporated into the local training objective to guide PretrainedBertMetaNet toward producing privacy parameters that comply with a predefined budget.

After completing local updates, each client records the effective noise scale $\sigma_t$ applied during DP-SGD and reports it to the server. The server aggregates the received updates:

$$\theta^{(t)} \leftarrow \sum_{i \in S_t} \frac{n_i}{\sum_{j \in S_t} n_j} \theta_i^{(t)}. \tag{19}$$

Subsequently, the server performs RDP accounting to estimate the cumulative privacy budget up to the current round $t$. This cumulative budget is then fed back to the clients participating in the next round as a penalty signal, guiding MetaNet toward budget-aware parameter adaptation.

## A.2 THE USE OF LARGE LANGUAGE MODELS (LLMs)

We disclose that large language models were used solely for text polishing and grammar refinement. All technical ideas, methods, experiments, and results are entirely the work of the authors, who take full responsibility for the content of this paper.

