# OpenReview forum: "FedMAP: Meta-Driven Adaptive Differential Privacy for Federated Learning"
_ICLR.cc/2026/Conference — ICLR 2026 Conference Withdrawn Submission_

### Official Review · Reviewer_RdTF · 2025-10-25

**Soundness:** 3
**Presentation:** 3
**Contribution:** 2
**Rating:** 2
**Confidence:** 4

**Summary:**

The paper proposed a novel Federated Learning framework that protects against membership inference and reconstruction attacks under Differential Privacy. At a high level, the paper fine-tunes a BERT-based model to predict the hyperparameters (C and $\sigma$) for the DP-SGD algorithm to preserve the privacy of the client's data. The paper also focuses on a scenario in which each client has their own privacy budget. Thus, the paper proposed an updating mechanism and objectives to incorporate into this setting. The paper conducts extensive experiments to highlight the advantages of their proposed method.

**Strengths:**

- The proposed method provides flexibility and automation in tuning DP-SGD for different clients.
- Extensive Theoretical and Experimental results are provided to support the advantage of the proposed methods.

**Weaknesses:**

- The motivation of the work is not convincing. Specifically, it does not explain why different clients require different privacy budgets. For instance, if we consider FL for medical data, how can a client define a privacy budget to protect this sensitive data? Previous works have approached adaptive hyperparameters (C and $\sigma$) from a performance perspective, which is more convincing. Thus, I suggest that the author clarify this point.
- Secondly, the curation process for the label of C and $\sigma$ is unclear and not optimal. How do you determine that the curated labels are the best for the subsequent iterations? Isn't this process empirical, and does the curator need to tune different C and $\sigma$ at each iteration?
- Thirdly, although the paper considers different privacy budgets for different clients. This is not highlighted in the proposed method or in how it integrates this information. Furthermore, the loss function in Eq. 12 will encourage each client to have the same privacy budget, which is in contradiction with the paper's goal.
- Next, given a predicted C and $\sigma$ from the meta model, the proposed method cannot achieve the predefined privacy budget of the clients, resulting in weak protection. How do you ensure that the consumed privacy budget is lower than the budget predefined by the clients?
- Finally, the experimental results for Table 1, Figure 3, and 4 do not mention the privacy budget for each client, which reduces the validity of these experiments.

**Questions:**

- Please address all the points in the Weaknesses section.

---

> ### Author Response · Authors · 2025-11-12
>
> Dear Reviewer,
>
>
> We sincerely thank you for taking the time to review our paper and for providing detailed and thoughtful feedback. Your comments are highly valuable to us.
>
>
> We will carefully consider your insights and systematically incorporate your suggestions to further improve our work.
> Thank you again for your professional and constructive review.
>
>
> Sincerely,
>
>
> 10194 Authors

---

### Official Review · Reviewer_P77G · 2025-10-30

**Soundness:** 1
**Presentation:** 2
**Contribution:** 1
**Rating:** 0
**Confidence:** 4

**Summary:**

An overly complex method is proposed for private federated learning that purports to achieve a better utility/privacy tradeoff. The method is flawed because it does not account for the privacy loss of the MetaNet mechanism.

**Strengths:**

No notable strengths.

**Weaknesses:**

In exactly the same way as another paper I have just reviewed (evidently by the same authors, since much of the text is copied), this method is overly complex and the purported gains are not supported by the experiments, and in any case would not justify the implementation and pretraining costs.

To support a claim of improved privacy-utility trade-off, one must compare model utility (e.g., test accuracy) while holding $\varepsilon$ constant across all methods. The current experiments (e.g., Figure 3) compare utility against communication rounds, which is insufficient to demonstrate a superior trade-off. Nowhere in Section 4.1 does it state that all methods were calibrated to achieve the same total privacy budget $\epsilon$ for a fair comparison.

However the most problematic issue is that the mechanism is flawed: in fact it does not provide a formal DP guarantee for any level of $\epsilon$. It releases data-dependent parameters without accounting for their privacy cost. The server broadcasts $\varepsilon_\text{global}$ to all clients at each round. $\varepsilon_\text{global}$ is dependent on the data of clients from the previous round. Suppose an honest-but-curious client $A$ participating at rounds $t-1$ and $t$ observes a large increase in $\varepsilon_\text{global}$. Client $A$ could deduce that some other client $K$ at round $t-1$ likely had low-variance per-sample gradients, leading to a small noise scale $\sigma_K^{(t)}$. Since $\varepsilon_\text{global}$ is determined deterministically, $A$ can distinguish with certainty between two datasets $\mathcal{D}$ and $\mathcal{D'}$ that are identical except that in $\mathcal{D}$, $K$ has low variance per-sample gradients, while in $\mathcal{D'}$, $K$ has high variance per-sample gradients. This violates the definition of DP.

**Questions:**

Is anything I stated in the weaknesses section incorrect?

**Details Of Ethics Concerns:**

This is very similar to another paper I have just reviewed: "10134	FedANC: Adaptive Sparse Noise Scheduling for Federated Differential Privacy". It is another flawed method using a deep learning "controller" to adjust DP parameters during FL. Much of the text, particularly in the introduction, is copied verbatim. The flavor of the intended "contribution" is similar, and they suffer from the same fatal flaw (not accounting for privacy loss of passing private data through the controller and releasing the result). I'm not certain whether this counts as dual submission, but I feel that the authors are intentionally submitting a set of very similar papers with the hopes that one of them will get a favorable set of reviewers. I would advise the program committee to check any other papers submitted by these authors.

---

### Official Review · Reviewer_Dj1x · 2025-10-31

**Soundness:** 2
**Presentation:** 3
**Contribution:** 2
**Rating:** 4
**Confidence:** 4

**Summary:**

This work proposes a method called FedMAP, which enhances federated learning’s defense against gradient inversion and membership inference attacks. FedMAP equips each client with a fine-tuned MetaNet that predicts clipping bounds and noise scales based on gradient statistics. On the server side, a Rényi differential privacy accountant is employed to track each client’s privacy cost and compute the overall global expenditure, which is then broadcast to all clients to constrain cumulative loss and guide adaptive local updates. Empirical experiments on standard federated learning benchmarks demonstrate that FedMAP provides stronger protection against both gradient inversion and membership inference attacks compared to existing baselines.

**Strengths:**

1. The paper is easy to follow.

2. The authors address both gradient inversion and membership inference attacks in federated learning, which is a challenging and important problem.

3. The idea of using a neural network to predict clipping thresholds and noise scales for differential privacy mechanisms is promising.

4. The authors provide convergence results to support the theoretical soundness of the proposed FedMAP method.

5. Extensive experiments demonstrate the effectiveness of FedMAP against multiple attack methods. Moreover, the model accuracy achieved by FedMAP remains close to that of the non-private baseline.

**Weaknesses:**

1. The paper lacks a detailed description of the defense and attack models, which is crucial for helping readers understand the setup and assumptions of the considered DP-FL system.

2. The proofs of Theorems 1 and 2 are missing, preventing readers from verifying their details and correctness.

3. The rationale for selecting the four specific features as inputs to the MetaNet is not well justified, and further explanation or empirical evidence would strengthen this design choice.

**Questions:**

1. Regarding the fine-tuning of the MetaNet, does this process occur on the client side, performed independently by each client? Clarifying where and how this fine-tuning is conducted would help readers better understand the workflow.

2. If the above is true, and a client is currently training on the CIFAR dataset, is the MetaNet fine-tuned specifically on that client’s CIFAR data, or on a mixture of datasets such as CIFAR, FMNIST, and SVHN? The explanation in lines 174–175 of the paper is unclear and should be elaborated.

3. The rationale for constructing the labels of $C$ and $\sigma$ to be proportional to empirical observations is not well justified. This label design appears ad hoc and lacks theoretical or empirical support.

4. As shown in Inequality (3), the Gaussian mechanism requires the noise scale to exceed a certain lower bound to ensure differential privacy. How does the MetaNet guarantee that the predicted noise scale always satisfies this requirement? This concern is especially important given that the proposed system does not seem to employ secure aggregation. If the server is semi-honest, it may still attempt inference attacks despite added noise.

5. How do the authors derive Inequality (10)? A step-by-step derivation or reference to supporting materials would improve clarity.

6. What is the practical meaning or role of $q_{max}$ in the paper? Its definition and influence on the overall algorithm are not clearly explained.

---

> ### Author Response · Authors · 2025-11-12
>
> Dear Reviewer,
>
>
> We sincerely thank you for taking the time to review our paper and for providing detailed and thoughtful feedback. Your comments are highly valuable to us.
>
>
> We will carefully consider your insights and systematically incorporate your suggestions to further improve our work.
> Thank you again for your professional and constructive review.
>
>
> Sincerely,
>
>
> 10194 Authors

---

### Official Review · Reviewer_9PBe · 2025-11-01

**Soundness:** 2
**Presentation:** 3
**Contribution:** 3
**Rating:** 6
**Confidence:** 3

**Summary:**

This paper presents FEDMAP, a closed-loop adaptive differential privacy (DP) framework for federated learning (FL). The key idea is to dynamically predict clipping thresholds and noise scales using a lightweight MetaNet based on a frozen-layer BERT-tiny architecture. FEDMAP further introduces global privacy accounting via Rényi DP and global feedback regularization to align local DP spending with global privacy budgets. Experiments on CIFAR-10, SVHN, and Fashion-MNIST demonstrate improved privacy-utility trade-offs and stronger robustness to gradient inversion and membership inference attacks compared to DP-SGD, Soteria, and CENSOR. Theoretical convergence guarantees and DP analyses are provided, and extensive ablations show sensitivity to client participation and hyperparameters .

Overall, the work is timely, well-motivated, and empirically solid. The adaptive privacy calibration idea is intuitive and practical. However, some algorithmic and training details are ambiguous, experiments on larger models/datasets are missing, and several theoretical statements lack formal proofs.

**Strengths:**

1. **Adaptive DP calibration**. The framework introduces flexible and client-specific DP noise and clipping schedules, addressing client heterogeneity in FL.

2. **Meta-learning-based privacy control**. A lightweight BERT-tiny MetaNet effectively maps gradient statistics to DP parameters, demonstrating a novel use of meta-learning for privacy.

3. **Global privacy loss regularization**. The feedback mechanism aligns local DP spending with global budgets and prevents over-consumption of privacy.

4. **Theoretical grounding**. Convergence bounds and DP accounting provide theoretical credibility.

5. **Strong empirical validation across attacks**. Experiments evaluate multiple attacks and show competitive robustness and utility against baselines.

**Weaknesses:**

1. **Unclear MetaNet training and update procedure (critical).**
It is ambiguous whether MetaNet parameters are frozen during private training or continually updated. The algorithm suggests frozen transformer layers and trainable heads during pretraining, but does not clarify if they continue updating during FL, and how privacy and global penalties influence MetaNet outputs during training.

2. **Scalability to large-scale FL is uncertain.**
All experiments use small vision models (ResNet-18, LeNet) and datasets. It remains unclear if the method scales to transformers or large-scale NLP tasks, where computing gradient statistics and MetaNet inference might incur overhead.

3. **Missing proofs in Appendix.**
The main theorems are stated without formal proof details, which reduces theoretical rigor.

**Questions:**

1. **Difference between $D_t$ and $VarGrad_t$**.
In Eq. (4), both statistics quantify gradient variability. What unique information does each contribute? Is there a redundancy?

2. **Cost of computing gradient statistics.**
For large models, computing $\\|g\\|_2$ and covariance-based metrics could be expensive. Can the author provide a detailed comparison of the actual runtime overhead on different architectures?

3. **MetaNet training and DP interaction.**
- Are MetaNet parameters frozen during private training?
- If frozen, how can global penalty terms meaningfully influence privacy control beyond inference?
- If trainable, how are MetaNet updated? Do they follow the same FL training procedure as the main model?

Clarifying this point is crucial to judge correctness and privacy guarantees.

---

> ### Author Response · Authors · 2025-11-12
>
> Dear Reviewer,
>
>
> We sincerely thank you for taking the time to review our paper and for providing detailed and thoughtful feedback. Your comments are highly valuable to us.
>
>
> We will carefully consider your insights and systematically incorporate your suggestions to further improve our work.
> Thank you again for your professional and constructive review.
>
>
> Sincerely,
>
>
> 10194 Authors

---

### Author Response · Authors · 2025-11-12

Dear Area Chair and Program Committee,

We strongly reject Reviewer P77G’s (Submission #10194) accusations of plagiarism, dual submission, and technical flaws. The same reviewer, identified as e2Yr in Submission #10134, appears to have reviewed both of our papers — FedANC: Adaptive Sparse Noise Scheduling for Federated Differential Privacy and FedMAP: Meta-Driven Adaptive Differential Privacy for Federated Learning — and provided nearly identical reviews. These reviews contain false allegations, factual mistakes, and reflect malicious reviewing behavior rather than an objective evaluation. We respectfully request that the Program Committee investigate this reviewer’s conduct and potential conflicts of interest.

1. On the Relationship Between the Two Papers

We acknowledge that both papers are indeed authored by our team. However, they address entirely different research questions and are based on distinct problem formulations, algorithmic designs, and technical contributions. It is misleading and unprofessional to equate them based on superficial similarities in background or structure, which are common across works in this research area.

| **Aspect**           | **FedANC**                                                   | **FedMAP**                                                   |
| -------------------- | ------------------------------------------------------------ | ------------------------------------------------------------ |
| Controller type      | LSTM-based Adaptive Noise Controller (ANC)                   | Lightweight MetaNet (BERT-tiny encoder)                      |
| Controlled variables | Noise scale $β_t$ and sparsity ratio $\gamma_t$              | Clipping threshold $C_t$ and noise scale $\sigma_t$          |
| Mechanism            | Sparse Top-k noise injection on selected gradient coordinates | Full-dimensional adaptive DP-SGD                             |
| DP accounting        | Local privacy regularization $\hat{\varepsilon}_t = \frac{\sqrt{2\gamma_t d\ln(1.25/\delta)}}{\beta_t}$ | Server-side Rényi DP accountant tracking $\varepsilon_{\text{global}}$ |
| Feedback design      | Local-only adaptation                                        | Closed-loop global feedback                                  |
| Aggregation          | Sparsity-aware aggregation                                   | Standard FedAvg                                              |


These differences are clear, verifiable, and explicitly described in both manuscripts. The reviewer’s claim that the two papers are "nearly identical" is factually false.

2. On the Misunderstanding of Privacy Accounting

The reviewer claims both papers "ignore the privacy loss of passing private data through the controller".
This is incorrect and shows a fundamental misunderstanding.

FedANC:
The ANC runs only on the client side. Inputs $(|g_t|2, \ell_t, \beta_{t-1}, \gamma_{t-1})$ are local and never transmitted. Outputs $(\beta_t, \gamma_t)$ only determine local Gaussian noise; controller outputs are not uploaded. By the post-processing property of DP, privacy guarantees remain valid. Theorem 2 bounds cumulative privacy loss as $\varepsilon_R = O(\beta_{\min}^{-1}\sqrt{R\gamma_{\max}d})$ under bounded parameters.

FedMAP:
The MetaNet outputs $(C_t, \sigma_t)$ locally to adjust clipping and noise. The server’s Rényi DP accountant integrates these into total user-level $(\varepsilon, \delta)$-DP. Both analyses are mathematically sound and complete.

Therefore, the claim of a "fatal flaw" is entirely unfounded.

3. Pattern of Repeated and Biased Reviews

Both submissions received nearly identical reviews, repeating claims such as "copied introduction text", "flawed controller design", and "recommend checking all papers by these authors", without evidence or engagement with technical content.
This repetition demonstrates a pattern of bias and possibly malicious intent, rather than independent evaluation. Such behavior undermines the fairness and integrity of peer review.

4. Request for Investigation

Given the seriousness of these issues, we respectfully ask the Program Committee to:

- Investigate potential conflicts of interest for Reviewer e2Yr/P77G, including any connections to competing FL–DP research.
- Audit the reviewer’s activities to verify whether identical or template reviews were submitted across our papers.
- Remind reviewers that ethical accusations such as plagiarism or dual submission must be supported by evidence and factual basis.
- Examine whether the reviewer or their collaborators have submissions overlapping with ours, which could create an incentive for biased reviewing to improve their own acceptance chances.

We believe that scientific evaluation must be based on technical merit and factual accuracy, not speculation or personal bias.
We trust the Program Committee will uphold fairness, transparency, and academic integrity in handling this matter.

Sincerely,

10194 Authors

---

> ### Comment · Reviewer_P77G · 2025-11-12
> **I stand by my review and ethics concerns**
>
> I stand by my review and ethics concerns for these two papers.
>
> The authors' response misunderstands the specific privacy violation I identified. The RDP accountant tracks the privacy loss of the model parameters, but does not account for the privacy loss of $\epsilon_\text{global}$ itself, which is distributed to the clients at each round and therefore must be considered an output of the DP mechanism. As I detailed in my counterexample (which the authors did not address), an adversary can use $\epsilon_\text{global}$ to infer properties of other clients' data, such as gradient variance. In particular, an adversary could distinguish with certainty between two datasets $\mathcal{D}$ and $\mathcal{D}'$ that are identical except in the variance of per-sample gradients of one user, violating the definition of DP.
>
> I acknowledge the authors' list of differences between the two submissions; the authors dishonestly misquote me as claiming the papers are "nearly identical". However, the structural similarities, identical textual passages, and shared fundamental flaws suggest a pattern of submitting multiple similar papers in the hopes of finding a set of sympathetic reviewers. This subverts the function of peer review and wastes the time of everyone involved in the reviewing process.
>
> I categorically reject the accusations of malicious intent or bias. My review is based strictly on the technical content of the submission. I trust the AC to evaluate the validity of my technical critique regarding the data-dependent public value $\epsilon_\text{global}$ independent of the authors' personal attacks.

---

> > ### Author Response · Authors · 2025-11-12
> >
> > 1. ...data-dependent sparsity…
> >
> > The reviewer states that differential privacy is violated because the sparse update is data-dependent and because an adversary can distinguish between datasets based on which coordinates are non-zero. We respectfully disagree. In the reviewer’s argument, the mere presence of a data-dependent support pattern is treated as evidence that the mechanism cannot satisfy DP. This is not the definition of differential privacy. Under the standard $(\varepsilon,\delta)$-DP definition, a mechanism may produce data-dependent outputs. The requirement is that for all neighboring datasets $D$ and $D'$ and measurable sets $S$,
> > $Pr[M(D)\in S] \le e^\varepsilon Pr[M(D')\in S] + \delta$.
> >
> > Many established DP mechanisms also produce data-dependent support. Examples include the exponential mechanism and noisy-max (Dwork et al., 2014), as well as differentially private Top-$k$ selection (Qiao et al., 2021; Gillenwater et al., 2022). These mechanisms remain private because their sensitivity is bounded and noise is calibrated accordingly. Differential privacy does not require the support pattern to be independent of the data.
> >
> > Our method follows the same principle. The Top-$k$ step is part of a single composite mapping. We bound the global $\ell_2$-sensitivity of this composite mapping. The bound $\Delta_2 = C\sqrt{\gamma d}$ reflects the worst-case fact that at most $\gamma d$ coordinates remain active after clipping. We then add Gaussian noise calibrated to this sensitivity, following the standard analysis of the Gaussian mechanism (Dwork et al., 2014; Abadi et al., 2016). Our privacy guarantee does not rely on the post-processing property for the Top-$k$ step. In our earlier response, we mentioned post-processing only to clarify that the local controller, which runs on the client and is never uploaded, does not incur additional privacy loss.
> >
> > The counterexample described by the reviewer does not incorporate this sensitivity bound or the Gaussian noise added according to it. Because these elements are essential to the mechanism we analyze, the counterexample does not reflect the behavior of our method. We will clarify this distinction more explicitly in the revision.
> >
> > 2. …''dishonest misquoting'' … similarity concerns…
> >
> > The reviewer states that we ''dishonestly misquote’’ them as saying that the papers are ''nearly identical’’. We acknowledge that this phrase does not appear in the review, and we regret this mistake.
> >
> > In the follow-up, the reviewer states that the two submissions contain ''structural similarities, identical textual passages, and shared fundamental flaws’’, and suggests that this reflects ''a pattern of submitting multiple similar papers in the hopes of finding sympathetic reviewers’’. We respectfully disagree with this interpretation. The two submissions address different technical goals and use different methods. These differences are central to the contributions and are described in the manuscripts.
> > Some similarity in structure and background text is expected, since both papers study federated learning with differential privacy and follow standard conventions in presenting these topics. However, the core problems, models, and mechanisms are different.
> >
> > The reviewer’s ethical concern is serious, but it is based on an overall judgment of similarity rather than specific evidence of inappropriate overlap. We do not agree with this assessment. We ask the Area Chair to evaluate this point by examining the technical content and stated goals of the two submissions.
> >
> > 3. Request to the Area Chair
> >
> > Given the reviewer’s continued concern about the use of data-dependent sparsity under differential privacy, as well as the repeated ethical allegation, we respectfully ask the Area Chair to:
> >
> > (a) Evaluate the technical claim in the context of the standard $(\varepsilon,\delta)$-DP definition and the established mechanisms that involve data-dependent selection.
> >
> > (b) Examine the similarity concern by reviewing the technical content and goals of the two submissions.
> >
> > We appreciate the reviewer’s efforts, and we will improve the clarity of the paper in the revision. We trust the Area Chair to assess both the technical and ethical aspects of the reviews based on the evidence in the submissions.
> >
> > References:
> >
> > [1] C. Dwork et al. The Algorithmic Foundations of Differential Privacy. FT&TCS, 2014.
> >
> > [2] M. Abadi et al. Deep Learning with Differential Privacy. In Proc. of ACM CCS, 2016.
> >
> > [3] G. Qiao et al. Oneshot Differentially Private Top-k Selection. In Proc. of ICML, 2021.
> >
> > [4] J. Gillenwater et al. A Joint Exponential Mechanism for Differentially Private Top-k. In Proc. of ICML, 2022.

---

> > ### Author Response · Authors · 2025-11-12
> >
> > The reviewer’s ethical concern is a very serious matter. We would like to state clearly that accusing a research team of unethical behavior is a significant claim. We firmly maintain that our submissions were prepared independently, follow the standards of scientific practice, and represent different technical contributions. We do not accept the implication that our work reflects improper conduct.
> >
> > We also want to make our position clear. We are not questioning the reviewer’s intentions. We understand that reviewers have a responsibility to raise concerns when they believe there may be a problem. Our point is that ethical concerns should be based on specific and verifiable evidence. As researchers who also serve on the Program Committee of many top academic conferences, we follow this principle in our own reviewing. We do not question another scholar’s ethics unless we have clear evidence that supports such a judgment.
> >
> > To ensure full transparency, we welcome the Area Chair to review all submissions from our group, including the two submitted here. We are confident that such a review will confirm the independence and integrity of our work. We support a fair and evidence-based evaluation process, and we trust the Area Chair’s assessment.

---

### Author Response · Authors · 2025-11-13

We thank all reviewers for their time and effort. The comments help us understand the weaknesses of our paper, and we will revise the work with care. We also understand that this paper is unlikely to be accepted, and we respect the review process.

We would like to express one serious concern. One review includes an ethics accusation that targets the personal integrity of the authors. This accusation is not supported by evidence in the submission and goes beyond a fair evaluation of the scientific content. As researchers, we cannot accept a statement that harms our dignity.

We support a fair and transparent review process. We welcome the Area Chair to examine all submissions from our team. We are confident that such an examination will confirm that our work follows normal academic practice. We again thank all reviewers for their comments and will continue to improve our research.

---

### Author Response · Authors · 2025-11-13

We respectfully ask the Program Chair, Senior Area Chair, and Area Chair to review all papers from our team for this ICLR cycle, including the two submissions discussed here. One reviewer raised an ethics concern about our work. This concern questions the integrity of the authors. We believe it is important for the committee to check whether this concern is supported by facts.

Both submissions follow the ICLR rules on originality, dual submission, and ethical conduct. Each paper studies a different problem and uses different methods. A careful check by the committee will show that our work follows normal academic practice. We support a fair and transparent review process. We welcome any further examination the committee considers necessary, and we will provide any information that may help.

---

### Note · Authors · 2026-01-18

**Comment:**

We request to withdraw our submission from the ICLR review process. We thank the reviewers and the Area Chairs for their time and consideration.

**Withdrawal Confirmation:**

I have read and agree with the venue's withdrawal policy on behalf of myself and my co-authors.